# A Case for Offering HPV Self-Sampling to Well-Screened Women. Comment on Lesack et al. Willingness to Self-Collect a Sample for HPV-Based Cervical Cancer Screening in a Well-Screened Cohort: HPV FOCAL Survey Results. *Curr. Oncol.* 2022, 29, 3860–3869

**Roni Kraut** [1,*], **Donna Manca** [1], **Aisha Lofters** [2] **and Oksana Babenko** [1]

1   Department of Family Medicine, University of Alberta, Edmonton, AB T6G 2T4, Canada; dpmanca@ualberta.ca (D.M.); oksana.babenko@ualberta.ca (O.B.)
2   Department of Family and Community Medicine, University of Toronto, Toronto, ON M5G 1V7, Canada; aisha.lofters@utoronto.ca
*   Correspondence: rkraut@ualberta.ca

**Abstract:** Lesack et al. recently published a cross-sectional study that focused on human papillomavirus (HPV) self-sampling in the screened population, a population not conventionally thought of for HPV self-sampling. They found 52% of well-screened, highly educated women who participated in the Human Papillomavirus For Cervical Cancer (HPV FOCAL) screening trial in British Columbia, Canada, would be willing to self-collect an HPV sample. We published a similar study in 2021 on well-screened, highly educated women affiliated with a family medicine clinic in Edmonton, Alberta, Canada, and found that 60% of these women preferred to have the option of HPV self-sampling. Our findings reinforce Lesack et al.'s results and together provide evidence for offering HPV self-sampling as an option for the well-screened population.

**Keywords:** human papillomavirus; HPV; HPV self-sampling; HPV testing for cervix screening; attitudes and acceptance toward HPV self-sampling

We read with interest the recently published study by Lesack et al. "Willingness to Self-Collect a Sample for HPV-Based Cervical Cancer Screening in a Well-Screened Cohort: HPV FOCAL Survey Results [1]".

We published a similar, albeit smaller, study in the Journal of Lower Genital Tract Disease in 2021 [2]. Our study provides evidence in support of the findings in Lesack et al.'s study.

We invited female patients (*n* = 212) in a family medicine clinic in Edmonton, Alberta, to answer a survey on their attitudes toward human papillomavirus (HPV) self-sampling as they were waiting for their appointment. Similar to Lesack et al.'s study, our survey included a brief description of self-sampling; however, we also included an illustration of the self-sampling process. Ninety-four percent of women took the survey, and, as in Lesack et al., these women were highly educated.

Lesack et al. found that 52% of women agreed or strongly agreed with the statement "I would be willing to collect my own sample/specimen for cervical cancer screening". In our study, we asked women a slightly different set of questions: (1) Whether they should have the option for HPV self-sampling and (2) Which option (HPV self-sampling, Pap, or undecided) they would select for future cervical cancer screening and why. We found that 60% of women agreed or strongly agreed with having the option of HPV self-sampling available to them, and 54% of women would select a Pap test, 24% of women would select HPV self-sampling, and 22% of women were undecided. Women who selected the Pap test had concerns that centered on taking an HPV self-sample correctly and the accuracy of HPV self-sampling. Given these reasons, it is likely a greater percentage of women would

have selected HPV self-sampling had they been given more than a brief description of the process as well as its accuracy.

In addition, Lesack et al. found a significant association between a higher level of education and willingness for HPV self-sampling. Our multivariate analysis did not find a significant association for this variable but did find a trend toward significance (odds ratio 2.3 [95% confidence interval: 0.56–9.41]); the lack of significance could be due to the smaller sample size in our study.

The similarity in findings between our study and Lesack et al.'s study is especially noteworthy given that the studies were conducted in different geographic locations and independently of one another. We hope that these synergistic findings will encourage policy makers to consider HPV self-sampling as an option for well-screened women in their cervical cancer screening programs.

**Author Contributions:** Conceptualization, R.K.; writing—original draft preparation, R.K.; writing—review and editing, R.K., D.M., A.L. and O.B. All authors have read and agreed to the published version of the manuscript.

**Funding:** This research received no external funding.

**Conflicts of Interest:** The authors declare no conflict of interest.

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
