# Peer review of "A Case for Offering HPV Self-Sampling to Well-Screened Women. Comment on Lesack et al. Willingness to Self-Collect a Sample for HPV-Based Cervical Cancer Screening in a Well-Screened Cohort: HPV FOCAL Survey Results. Curr. Oncol. 2022, 29, 3860–3869"

_curroncol, doi:10.3390/curroncol29080425_

Round 1

Reviewer 1 Report

I believe that this commentary is appropriate and does add additional value to the primary manuscript.

Reviewer 2 Report

The authors present Comment and draw attention to their earlier smaller study by Kraut et al. (2021) in connection with the recently published study by Lesack et al. (2022). Both studies focused on the willingness of highly educated women to self-collect a sample for HPV screening. The studies differ in the number of participants (212 vs. 4,945, respectively), geographic area, and survey questions. In the submitted Comment, the authors should add other characteristics of the patient group (age, education, etc.) that may influence the study results and discuss.